# PRIVATE RETRIEVAL AUGMENTED GENERATION WITH RANDOM PROJECTION

**Dixi Yao, Tian Li**
Department of Computer Science
University of Chicago
Chicago, IL 60637, USA
{dixi,litian}@uchicago.edu

## ABSTRACT

Large Language Models (LLMs) have gained widespread interest and driven advancements across various fields. Retrieval-Augmented Generation (RAG) enables LLMs to incorporate domain-specific knowledge in a non-parametric way. However, evidence shows that prompting retrieval-augmented models can pose significant privacy risks due to leakage of sensitive information stored in the retrieval database. In this work, we propose a private randomized mechanism to project both the queries and the datastore into a lower-dimensional space using Gaussian matrices, while preserving similarities for effective retrieval. Empirical evaluation on different RAG architectures shows that our solution achieves strong empirical privacy protection with negligible impact on generation performance and latency compared to prior methods.

## 1 INTRODUCTION

Large Language Models (LLMs) excel across diverse applications. Retrieval-Augmented Generation (RAG) (Khandelwal et al., 2019; Lewis et al., 2020; Min et al., 2023; Edge et al., 2024) enhances LLMs by enabling them to answer domain-specific questions using external knowledge without additional training, ensuring easy deployment. However, recent studies reveal that sensitive information in the knowledge database is vulnerable to leakage if attackers craft specific prompts (Huang et al., 2023; Zeng et al., 2024b; Koga et al., 2024). For example, a company's user data, such as emails and names, stored in RAG documents, can be exposed through API interactions.

Despite the growing adoption of RAG and the simplicity of these attacks, effective countermeasures are lacking. Attacks mimic ordinary user queries, and simple access control may block legitimate users. Following the definition of record-level differential privacy (DP) (Dwork et al., 2006) (where each entry in the datastore is one record), our goal is to mask the information stored in the database so that the model outputs are close to those in non-private settings but sensitive information like personal emails and phone numbers is blurred.

To this end, we propose a novel privacy-preserving solution for RAG via random projection. By projecting the private database onto a permuted space, we ensure answers exclude sensitive information while retaining essential content. Our key idea is that random projection preserves pairwise similarity in the lower-dimensional space, yet changes record embedding values such that attackers extract alternative items instead of the original text. This allows users to receive approximately accurate answers from the RAG system while adversaries may fail.

This paper considers two popular RAG scenarios: KNN-LM (Khandelwal et al., 2019) and direct prompting with retrieval outputs (Min et al., 2023). KNN-LM enhances next-word prediction by combining LLM probabilities with those from retrieved nearest-neighbor entries. Direct prompting feeds the retrieved items together with the query into the LLM to generate responses.

Empirical results show that our method outperforms prior work and direct random projection (Li & Li, 2023), where data embeddings are projected by a matrix of IID random variables from a standard Gaussian distribution, which is then combined with another Gaussian random matrix to

ensure differential privacy. Our approach achieves better generation quality and datastore privacy, offering a promising direction for private RAG.

## 2 RELATED WORK

**Retrieval Augmented Generation.** Retrieval-Augmented Generation (RAG) enhances pre-trained LLMs with non-parametric retrieval of external knowledge. Retrieval methods based on $k$-nearest neighbors (Khandelwal et al., 2019) retrieve relevant database entries, combined with LLMs to produce the final answer. In KNN-LM, the next-word probability is obtained by linearly combining language model predictions with probabilities from retrieved items. Another RAG architecture (Lewis et al., 2020; Min et al., 2023) directly prompts LLMs with a concatenation of retrieved items and the original query to generate final responses. We call such methods *direct-prompting RAG* in this paper. For instance, GraphRAG (Edge et al., 2024) builds a graph-based text index and uses it to retrieve from the knowledge base.

**Privacy of RAG.** LLMs can pose privacy risks due to strong memorization of information. Previous works show that personal data (e.g., emails, phone numbers, and URLs) and random datastore content can be extracted from KNN-LM models (Huang et al., 2023). Zeng et al. and Jiang et al. confirm similar vulnerabilities in RAG models via direct prompting. Efforts to defend such attacks include DP-based sampling and aggregation (Koga et al., 2024) over tokens and usage of synthetic data (Zeng et al., 2024a). However, these methods reduce efficiency, particularly with large datasets, requiring extra validation data or additional training.

**Random Projection.** Random projection reduces dimensionality while preserving pairwise distances with high probability. By projecting query and datastore embeddings to lower dimensions, retrieval accuracy remains largely unaffected (Section 4). Additionally, Blocki et al. (Blocki et al., 2012) prove that random projection with a standard random Gaussian matrix satisfies differential privacy (see Definition 1). Prior work applied DP random projection to aggregation (Li & Li, 2023) and image retrieval (Ibrahim et al., 2024). Instead of directly applying random projection after embeddings in RAG, we propose an algorithm involving matrix projection to perturb data embeddings for both KNN-LM and direct-prompting RAG.

**Definition 1** (Differential Privacy (DP) (Mironov, 2017)). *A randomized mechanism $\mathcal{M} : \mathcal{D} \to \mathcal{S}$ satisfies $(\epsilon, \delta)$-differential privacy if for any neighboring datasets $D_1$ and $D_2$ differing by one record, it holds that*

$$\Pr[\mathcal{M}(D_1) \in \mathcal{S}] \le e^\epsilon \Pr[\mathcal{M}(D_2) \in \mathcal{S}] + \delta.$$

In our context, we aim to achieve token-level DP (Huang et al., 2023; Koga et al., 2024). Let the neighbouring datasets be $D_1, D_2 \in \mathbb{R}^{n \times d}$, which differ by one token embedding (row). Token-level privacy guarantees that the output of the RAG architecture would follow similar distributions under the two datasets. In other words, with or without the presence of some sensitive word embedding, the output of our random algorithm would be similar.

## 3 METHOD

Our proposed algorithm is shown in Algorithm 1. In the first step, we choose a value of random Gaussian noise to be used in the random projection. In Line 4, we only pick the first $n$ entries of the document $D$ so that the document size is fixed. A pre-trained language model $f(\cdot)$ serves as the encoder, encoding $D = \{w_1, \cdots, w_n\}$ into $f(D) = \{f(w_1), \cdots, f(w_n)\}$ (Line 4), which can be further finetuned. But we assume a fixed pre-trained LM throughout the paper for simplicity. Each embedding $w_i$ ($i \in [n]$) has a dimension of $d$.

We then randomly generate an IID Gaussian Matrix $R \in \mathbb{R}^{d \times k}$, where each cell is sampled from $\mathcal{N}(0, \sigma^2)$. Next, after normalizing (or clipping) $D$, we project $f(D)$ to $f(D)R$ (Line 4). To preserve similarities between queries and datastores, we project the input query $x$ to the same dimension of $k$, using the same matrix $R$. Then, we calculate the distances using the projected embeddings of the query and dataset for the nearest neighbor search. The following steps are the same as in the common process reflected by calculation by $M(\cdot)$ in Line 7.

Our algorithm is compatible with both KNN-LM and direct prompting in RAG architectures. In KNN-LM, the next-word probability combines logits from a pre-trained LLM and softmax prob-

---

**Algorithm 1:** Private RAG with Random Projection

---

**Data:** Datastore $D$, Embedding encoder $f(\cdot)$, $M(x, f(D))$

1 . **Parameters** : Gaussian matrix magnitude $\sigma$, normalization bounds $\gamma$ and $\Delta$, projection dimension $k$, and max document entry $n$.

**Input:** An arbitrary query $x$.

**Output:** Next work prediction $y$.

2 *Run once before taking user queries:*

3 Generate a $d \times k$ IID random matrix from $\mathcal{N}(0, \sigma^2)$.

4 Pick the first $n$ entries of the document $D$, embed $D$ with some encoder $f(\cdot)$, and get $f(D)$.

5 Normalize and clip $f(D)$ so that each element $\gamma \leq \|f(w_i)\|_2 \leq \Delta, (1 \leq i \leq n)$. Project $f(D) \rightarrow f(D)R$.

6 **for** *each user query $x$:* **do**

7 $\quad$ $y \leftarrow M(xR, f(D)R)$

8 **end**

---

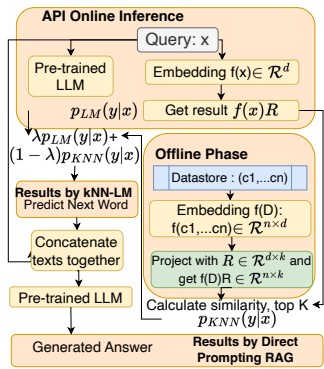

Figure 1: Working flow of KNN-LM and direct prompting RAG.

abilities based on the distance of token candidates to the nearest neighbor. In direct prompting, retrieved texts, such as those from KNN, are combined with queries and input into another LLM to generate a final answer. As illustrated in Fig. 1, the green block denotes our random projection operations, corresponding to line 5 in Algorithm 1. After retrieving embeddings from the datastore, we can either directly output the prediction, yielding results from KNN-LM architectures, or combine the query with the retrieved information and input it into another pre-trained LLM. In the other architecture, we prompt the LLM with the retrieved texts alongside the query to generate the result.

### 3.1 PRIVACY IMPLICATIONS

In this paper, we evaluate privacy through empirical attacks. However, in this section, we would like to provide a discussion on how our algorithm can achieve potentially differential privacy guarantees, and we leave a formal proof to the future version of the paper. For any datastore $D_1$, we generate an adjacent datastore $D_2$ by either adding a row or deleting any row from $D_1$. By definition and postprocessing properties of differential privacy, we would like to have the probabilities of $D_1R$ and $D_2R$ hard to differentiate. For instance, if the differing entry contains sensitive information (e.g., removing a sensitive entry in $D_1$), we can still generate similar answers based on $D_2$. However, as $D_2$ does not contain the entry, which potentially contains private information, an attacker may fail.

We note that the distribution of each element in $D_1R$ and $D_2R$ follows IID normal distributions and denote them as $\mathcal{N}(0, \sigma_1^2)$ and $\mathcal{N}(0, \sigma_2^2)$ respectively, where $\sigma_1^2$ is $\sigma^2\|D_{1_m}\|_2^2$ and $\sigma_2^2$ is $\sigma^2\|D_{2_m}\|_2^2$. By clipping each embedding vector, we can bound the difference of the two variance.

Here, we can consider several parameters which will affect the tradeoff between privacy and generation performance. Smaller projection dimension $k$ represents higher compression rate of information as we need to project features from dimensions of $d$ to $k$. Higher compression rates can implicitly impose stronger privacy while decreasing generation performance. Another factor we shall consider is the value of $\frac{\Delta^2}{\gamma^2}$, which is the ratio between the upper and lower clipping bounds. As this value is related to the variance of the distribution of projected matrices, it will also affect the privacy/utility tradeoffs. Apart from $k$ and clipping/normalization bounds, another parameter $\sigma$ can implicitly influence the tradeoff between resistance to extraction attacks and language generation performance. Setting $\sigma$ too high increases the input magnitude to the language model, potentially leading to inaccurate predictions.

## 4 EVALUATION

### 4.1 METRICS AND SETUP

To evaluate privacy of different approaches, we conduct empirical extraction attacks (Huang et al., 2023) on the RAG model using specific prompts to extract details (e.g., emails, websites, phone numbers). We use the Enron Email dataset (Klimt & Yang, 2004), which does not overlap with those used to pre-train major LLMs like GPT-2 (Radford et al., 2019), ensuring the RAG data are

| Methods | Email | URL | Phone Number | Perp. | Perp. -Sens |
|---|---|---|---|---|---|
| KNN-LM | 0 | 13 | 0 | 2.872 | 2.872 |
| Ours | 0 | 0 | 0 | 2.89 | **2.20** |
| DP-RP-G | 0 | 2 | 0 | 2.96 | 2.89 |

Table 1: KNN-LM

| Methods | Email | URL | Phone Number | Perp. | Perp. -Sens |
|---|---|---|---|---|---|
| RAG | 36 | 56 | 125 | 1.58 | 1.58 |
| Ours | 0 | 3 | 0 | 2.05 | **1.06** |
| DP-RP-G | 0 | 3 | 0 | 2.05 | 1.59 |

Table 2: Direct-prompting RAG

unseen by the pre-trained model. Thus, private data is limited to RAG documents. We used GPT-2 (Radford et al., 2019) as the pre-trained LLM. In KNN-LM, $\lambda$ in KNN-LM is 0.1 and $K$ in KNN is 1024. We set $\gamma = 1$, $k = 64$, $\Delta = 2$, and $\sigma = 0.1$. For Enron email dataset, the datastore size $n$ is 465026. For GPT-2, the model embedding size is 768.

We follow prior work (Huang et al., 2023) to report perplexity. Lower perplexity indicates higher generation quality. In addition, we also measure the perplexity where sensitive information is removed from the datastore. We call it perplexity-sens (or Perp.-Sens. in the table). The reason is that we do not hope to generate high-quality sensitive information. If privacy-preserving solutions work well, the generated content will not have sensitive information by design. Therefore, the generation of sensitive information should not be taken into account for a fair comparison.

## 4.2 RESULTS

To verify our algorithm's effectiveness, we compare it with baselines of non-private methods and directly applying DP random projections under different privacy budgets. Apart from performance and generation, we measure the latency for one RAG inference. Under the same setting, only using GPT-2 without any retrieval components yields a perplexity of 3.31, which is far from satisfactory.

### 4.2.1 KNN-LM

The results show that even under a tight privacy budget, generation performance is largely maintained with minimal loss. Interestingly, performance improves when sensitive information is excluded, as random projection masks overly detailed data, allowing KNN searches to focus on meaningful content rather than specifics like phone numbers. This leads to more contextually appropriate word selection and better outcomes. Directly applying random projection (Li & Li, 2023) over query and datastore embeddings requires privacy preservation at the cost of performance degradation, with two items still leaked. Averaging over 50,000 inferences, each inference takes KNN-LM 0.793s and our method 0.811s on Nvidia RTX 4090.

### 4.2.2 DIRECT-PROMPTING LM

Apart from KNN-LM, we apply our methods to modern RAG architecture. We first use KNN-LM to retrieve the text. We then put the retrieved content along with the query in a message template and input them into a pre-trained LLM for answers, following the paradigm of the prevalent RAG.

Compared to KNN-LM, modern RAG achieves better generation performance by combining retrieval results with queries for more aggressive generation. However, this increases privacy leakage, with over 100 personal phone numbers retrievable. Fortunately, our methods prevent such leaks while maintaining similar results and improving perplexity-sens. Perplexity worsens slightly as sensitive information is excluded from the generated results. In summary, our approach successfully defends against attacks without sacrificing generation performance. Each inference takes RAG 2.996s and our applied methods 2.975s.

## 5 DISCUSSION AND FUTURE WORK

This paper presents preliminary results to show that randomly projecting RAG document embeddings onto a lower-dimensional space and queries does not hurt the performance of RAG architectures, while can bring about privacy benefits validated through empirical attacks. In future work, we aim to further explore the formal privacy guarantees and apply our method to more advanced and complicated RAG models.

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
