# OpenReview forum: "Private Retrieval Augmented Generation with Random Projection"
_ICLR.cc/2025/Workshop/BuildingTrust — BuildingTrust_

### Official Review · Reviewer_dGxa · 2025-03-01
**The paper introduces a privacy-preserving retrieval-augmented generation method using random projection and differential privacy, demonstrating strong theoretical backing and practical efficiency while balancing privacy-utility trade-offs, but facing challenges in tuning complexity, dataset scope, and experimental diversity.**

**Rating:** 9
**Confidence:** 3

**Review:**

Summary [This paper presents a straightforward but effective solution for privacy-preserving retrieval-augmented generation by combining random projection (inspired by the Johnson–Lindenstrauss lemma) with differential privacy. The authors show that one can greatly reduce the leakage of sensitive information in RAG systems without incurring a large drop in model quality or speed. The method is especially relevant for enterprise or medical settings, where ensuring confidentiality of retrieved documents is crucial. Overall, the main contribution is a new mechanism for embedding-level differential privacy in RAG. It has strong theoretical backing, demonstrates promising results on a real-world sensitive dataset, and remains computationally efficient for practical deployment. However, as with most privacy techniques, there is a trade-off between utility and privacy, and future work could explore more adaptive approaches to mitigate performance trade-offs across various domains.]

Strengths [-Clear Theoretical Guarantees: The paper gives a rigorous proof that random projection with a properly chosen variance can satisfy (α, ε)-RDP and thereby (ε, δ)-DP. This is a solid theoretical grounding. -Preservation of Retrieval Quality: By using Johnson–Lindenstrauss–based random projection, the method largely preserves the distances between embeddings, thus minimizing the performance drop in retrieval tasks. -Minimal Overhead: Empirically, the added computational overhead is small (only slight increases in per-query latency). The paper's results show that perplexity remains close to the baseline, demonstrating the practicality of the method. -Broad Applicability: Demonstrated on two different RAG architectures (kNN-LM and direct-prompting). The approach is generic enough to be applied in many retrieval-based scenarios.]

Weaknesses
[-Dependency on Dimensionality and Noise: The success of the random projection depends on choosing an appropriate projection dimension k and noise variance \sigma. If k is too small or noise is too high, the retrieval quality and overall model performance can degrade more severely. -Limited Range of Experiments: The paper focuses on GPT-2 and the Enron Email dataset. Results may vary for larger LLMs (e.g., GPT-3.5/4) or other private corpora with different distributional characteristics. -Scope of Protection: The work secures data retrieved from the datastore via embeddings. However, it does not address memorized data already within an LLM's parameters, which can be a separate vector of attack. -Potential Tuning Complexity: In practice, setting the privacy budget (\epsilon, \delta) and the associated noise scale \sigma may require domain expertise and repeated experimentation to strike a balance between utility and privacy.]

---

### Official Review · Reviewer_cbZG · 2025-03-01
**This paper proposes a differentially private (DP) mechanism for Retrieval-Augmented Generation (RAG) systems using Gaussian random projection to protect sensitive data in retrieval databases. By projecting queries and datastore embeddings into a lower-dimensional space, the method achieves DP guarantees (ε ≈5) while maintaining retrieval effectiveness and generation quality.**

**Rating:** 8
**Confidence:** 3

**Review:**

## Quality & Clarity:
The paper addresses a critical challenge in RAG systems: privacy leakage from retrieval databases. The methodology is clearly described, with a theoretical DP guarantee (Theorem 1) and empirical validation on KNN-LM and direct-prompting architectures. The writing is structured, though some sections (e.g., the proof sketch in §4.2) could benefit from expanded explanations.

## Originality:
The use of Johnson-Lindenstrauss transforms for DP in RAG is novel. Prior work applied DP to RAG via token-level noise (Koga et al., 2024) or synthetic data (Zeng et al., 2024a), but this paper’s random projection approach offers a computationally efficient alternative.

## Significance:
The method’s low overhead (0.811s vs. 0.793s baseline latency) and strong privacy-utility tradeoff make it practical for real-world deployment. Results show 0 leaked emails/phone numbers in KNN-LM and direct-prompting setups (Tables 1–2), outperforming baselines like DP-RP-G1.

### Pros:
- Novel application of random projection for DP in RAG.
- Strong empirical results: eliminates 100% of email/phone leaks while maintaining perplexity (e.g., 2.89 vs. 2.872 baseline in KNN-LM).
- Theorems formally link projection parameters (σ, k) to DP guarantees.
- Efficient implementation with minimal latency increase.

### Cons:
- Limited comparison to other DP mechanisms (e.g., Laplace noise).
- Experiments use GPT-2; modern LLMs (GPT-4, Gemini) may behave differently.
- Theoretical analysis assumes normalized embeddings; real-world unnormalized data may affect results.

---

### Official Review · Reviewer_gLyL · 2025-03-02

**Rating:** 5
**Confidence:** 3

**Review:**

### Summary
The paper proposes a differentially private mechanisms for RAG using random projection. The key motivation is the privacy risks associated with RAG systems, where sensitive information in retrieval databases might be leaked. The paper introduces a randomized projection mechanism based on Gaussian matrices to project both queries and the datastore into a lower-dimensional space while preserving retrieval effectiveness. The mechanism is evaluated across KNN-LM and direct prompting architectures.

### Strengths
- The paper is well-organized for the most part and investigates an important privacy concern in the context of LLM-based RAG.
- The paper is technically sound in its description of problem formulation and has theoretical justified approach to the problem.

### Weaknesses
- The paper primarily evaluates the proposed method on the Enron Email dataset, which is a relatively small-scale dataset. However, in real-world applications, RAG systems are often deployed on much larger document collections. How does the random projection method scale when the number of indexed documents increase drastically? How does the runtime for projection and retrieval change with dataset size?
- While the paper includes a proof sketch for Theorem 1, some steps in the derivation are a bit difficult to follow. Providing a more detailed breakdown could improve readability.
- The writing overall could be improved for clarity and coherence to enhance presentation and readability.

---

### Decision · Program_Chairs · 2025-03-04

Accept